# Transient Global Ischemia-Induced Brain Inflammatory Cascades Attenuated by Targeted Temperature Management

**DOI:** 10.3390/ijms22105114

**Published:** 2021-05-12

**Authors:** Dae Ki Hong, Yoo Seok Park, Ji Sun Woo, Ju Hee Kim, Jin Ho Beom, Sung Phil Chung, Je Sung You, Sang Won Suh

**Affiliations:** 1Department of Physiology, College of Medicine, Hallym University, Chuncheon 24252, Korea; zxnm01220@gmail.com; 2Department of Emergency Medicine, Severance Hospital 50, College of Medicine, Yonsei University, Yonsei-ro, Seoul 03722, Korea; PYS0905@yuhs.ac (Y.S.P.); WANGTIGER@yuhs.ac (J.H.B.); 3Department of Emergency Medicine, Gangnam Severance Hospital 211, College of Medicine, Yonsei University, Eonju-ro, Seoul 06273, Korea; JISUNWOO224@yuhs.ac (J.S.W.); JUHEEKIM55@yuhs.ac (J.H.K.); emstar@yuhs.ac (S.P.C.)

**Keywords:** post cardiac arrest care, targeted temperature management, high mobility box protein 1, apoptosis, inflammation, microglia

## Abstract

Sudden cardiac arrest leads to a significantly increased risk of severe neurological impairment and higher mortality rates in survivors due to global brain tissue injury caused by prolonged whole-body ischemia and reperfusion. The brain undergoes various deleterious cascading events. Among these damaging mechanisms, neuroinflammation plays an especially crucial role in the exacerbation of brain damage. Clinical guidelines indicate that 33 °C and 36 °C are both beneficial for targeted temperature management (TTM) after cardiac arrest. To clarify the mechanistic relationship between TTM and inflammation in transient global ischemia (TGI) and determine whether 36 °C produces a neuroprotective effect comparable to 33 °C, we performed an experiment using a rat model. We found that TTM at 36 °C and at 33 °C attenuated neuronal cell death and apoptosis, with significant improvements in behavioral function that lasted for up to 72 h. TTM at 33 °C and 36 °C suppressed the propagation of inflammation including the release of high mobility group box 1 from damaged cells, the activation and polarization of the microglia, and the excessive release of activated microglia-induced inflammatory cytokines. There were equal neuroprotective effects for TTM at 36 °C and 33 °C. In addition, hypothermic complications and should be considered safe and effective after cardiac arrest.

## 1. Introduction

Sudden cardiac arrest (CA) is a leading cause of death in all countries [1,2]. In addition to a high risk of mortality, there is a significant risk of severe neurological impairment in CA survivors with the return of spontaneous circulation [2]. In the post-resuscitation phase after whole-body ischemia, the combined ischemic and reperfusion injuries and their underlying pathological process can cause death and neurological damage [2]. Although global tissue and organ injury are primarily caused by prolonged total body ischemia, additional damage occurs during and after reperfusion [3]. The mechanisms of brain injury are associated with altered calcium homeostasis, free radical formation, mitochondrial dysfunction, protease activation, inflammation, apoptosis, and necrosis [3,4,5,6].

Although existing therapies have limited efficacy in minimizing brain damage after CA [7], therapeutic hypothermia (sustained core temperature of 32 to 34 °C for 12 to 24 h) has shown significant improvement in neurologic outcomes and survival rates in resuscitated CA patients [8]. As all organ systems in the human body can be affected by clinically induced total body hypothermia, the application of therapeutic hypothermia (or some form of targeted temperature management (TTM)) should be considered due to its balanced risk versus benefit ratio [8]. With respect to survival and neurological benefit, one randomized trial published in 2013 compared patients treated with TTM at either 33 °C or 36 °C and demonstrated that the effects of TTM at 36 °C were just as beneficial as those at 33 °C [8]. The 2015 guidelines for Cardiopulmonary Resuscitation and Emergency Cardiovascular Care, which are derived from clinical trial results, suggest the selection and maintenance of a target core temperature between 32 °C and 36 °C for at least 24 h during TTM [1]. As TTM at 36 °C can decrease the risk of several adverse medical conditions, it may be as beneficial as TTM at 33 °C in resuscitated CA patients [9].

The application of TTM at 36 °C has been increasing worldwide. However, until now, it is unknown if TTM at 36 °C produces comparable neuroprotective effects against brain damage after CA with respect to molecular mechanisms.

The neuroprotective nature of TTM affects multiple aspects of brain physiology, including reactive oxygen species production, excitotoxicity, metabolic activity, blood flow, apoptotic pathways, inflammation, and blood–brain barrier integrity, as well as neurogenesis, gliogenesis, and angiogenesis in the acute, sub-acute, and chronic stages of injury [5,10]. Among these, inflammatory responses are thought to be the principal source of exacerbation of neurological damage after ischemic/reperfusion injury [5,6].

High mobility group box 1 (HMGB1), which is released from several types of cells in response to injury, acts as one of the primary mediators of this innate immune response, leading to severe sterile inflammation [11,12,13]. Following an ischemic event, HMGB1 is translocated to the intracellular fluid (ICF) and secreted into the extracellular space. Studies have indicated that intracellular HMGB1 plays an important role in the regulation of energy homeostasis and transcription. In contrast, it has been reported that extracellular HMGB1 directs blood–brain barrier breakdown, neuroimmune activities, and neuronal death [14]. HMGB1 is rapidly released from damaged cells within a short time after pathological conditions [12]. Extracellular-released HMGB1 acts as a danger signal by binding to toll-like receptor 4 proteins and activating inflammatory mediators, which in turn amplify and exacerbate brain damage [11,13].

Microglia, the resident immune cells of the central nervous system, are highly plastic [15]. Resting microglia can be activated into dual phenotypes (known as microglial polarization) under different microenvironment conditions [16]. After ischemic stroke, the biphasic functions of microglial polarization produce two types of microglia: the classical, detrimental, pro-inflammatory phenotype M1; and the alternative, beneficial, anti-inflammatory phenotype M2 [16]. A new approach in the suppression of microglial activation and the promotion of the M2 phenotype may provide new prospects for brain repair and new therapeutic strategies for stroke management [17]. However, no studies have investigated whether TTM can influence microglial activation and polarization [18].

HMGB1 may lead to the imbalanced activation and polarization of M1/M2 microglia. This imbalanced polarization of microglia may be avoided by blocking the release of HMGB1 [13,19,20]. As TTM can inhibit the extracellular release of HMGB1 after ischemia/reperfusion injury, it may be a potential modulator against microglial activation and polarization for the amelioration of sterile inflammation after CA.

Thus, in our study, we compared the neuroprotective effects of TTM at 33 °C and 36 °C after CA using a transient global ischemia (TGI) model. We hypothesized that TTM at both 33 °C and 36 °C may produce a robust, neuroprotective effect on ischemic injuries of the whole brain by preventing the release of HMGB1 from damaged cells, thereby inhibiting apoptosis, microglial activation, and imbalanced microglial polarization, as well as consequently diminishing the propagation of the inflammatory response.

## 2. Results

### 2.1. TTM at 33 °C and 36 °C Reduces Neuronal Cell Death and Improves Behavioral Function Following Transient Global Ischemia

To estimate whether TTM at 33 °C and 36 °C had a neuroprotective effect after TGI-induced neuronal death, degenerating neurons were evaluated in the hippocampal areas cornu ammonis 1 (CA1) and parietal cortex (PC) at 3 days after TGI. Degenerating neurons were visualized by using a fluorescent dye, Fluoro-Jade B (FJB), with a high specificity for degenerating apoptotic and necrotic neurons. When compared with the rats under normothermia (NT) at 37.5 °C, rats that received TTM at 33 °C and 36 °C after TGI showed dramatically reduced neuronal death in CA1 (Figure 1A). The neuronal cell death in TTM at 33 °C and 36 °C after TGI was approximately 68% less than that of NT at 37.5 °C (Figure 1B). In the adhesive removal test, there was no substantial improvement in removal time (120 s) in rats under NT at 37.5 °C, 3 to 72 h after TGI. However, rats under TTM at 33 °C and 36 °C showed significant improvement in removal time compared to sham rats after 24 h (Figure 1C). In addition, the TGI-induced neuronal death and behavioral test results between TTM at 33 °C and 36 °C did not differ significantly.

### 2.2. TTM at 33 °C and 36 °C Protect against Apoptosis-Mediated Cell Death Following Transient Global Ischemia

To demonstrate a relationship between TTM at 33 °C and 36 °C and apoptosis-mediated cell death, immunohistology assessments for terminal deoxynucleotidyl transferase dUTP nick end labeling (TUNEL), BCL2 Associated X, apoptosis regulator (BAX), and caspase-8 were performed in TGI-exposed brains of rats under NT at 37.5 °C and TTM at 33 °C and 36 °C. These immunostainings correlated with 4′,6-diamidino-2-phenylindole (DAPI) that is a blue-fluorescent DNA stain, ~20-fold enhancement of fluorescence upon binding to AT regions of double stranded DNA. At day 3 (72 h) after injury, rats under TTM at 33 °C and 36 °C were approximately 73% lower in TUNEL-positive cells than rats under NT at 37.5 °C. However, there was no significant difference in the number of TUNEL-positive cells between TTM at 33 °C and TTM at 36 °C (Figure 2A,B and Appendix A). Since BAX and caspase-8 are closely associated with apoptosis in ischemic injury, we assessed for changes in their expressions after the TGI in rats under NT at 37.5 °C, TTM at 33 °C, and TTM at 36 °C. On day 3 post-injury, the expressions of BAX and caspase-8 had increased significantly in the CA1 in rats under NT at 37.5 °C. Compared with the expression levels under NT at 37.5 °C, TTM at 33 °C and 36 °C reduced the expressions of caspase-8 and BAX expression by approximately 90 and 87% and approximately 88 and 94%, respectively (Figure 2C–F).

### 2.3. TTM at 33 °C and 36 °C Inhibits Extracellular Release of HMGB1 after TGI

HMGB1 is released from brain cell nuclei during TGI, thereby reducing the number of HMGB1-positive cells after ischemic injury. At 4 h after TGI, we found that HMGB1 immunoreactivity disappeared from the CA1 and PC; however, TTM at 33 °C and 36 °C significantly restored HMGB1-stained cells in both the CA1 and PC. We found that 10.58 ± 0.99% and 8.55 ± 0.76% of DAPI-positive cells in the CA1 and PC, respectively, were HMGB1-positive in the NT at 37.5 °C group. While TTM at 33 °C increases this number approximately 58.87 ± 1.26% and 57.17 ± 0.83% in CA1 and PC, respectively, TTM at 36 °C also increases this number approximately 60.31 ± 0.84% and 57.89 ± 0.90%, respectively. This suggests that TTM decreases the extracellular release of HMGB1 (Figure 3A–D). To compare the expressions of specific cytokines associated with the extracellular release of HMGB1 according to core temperatures, we conducted a real time polymerase chain reaction at 24 h after TGI using isolated mRNA with whole brain tissue. The quantitative polymerase chain reaction findings showed that the mRNA expression of TNF-α, IL-1β, and IL-6 decreased in the transient cerebral ischemic brain after TTM at 33 °C and 36 °C in comparison with NT at 37.5 °C (Figure 3E–G). An exception found that the mRNA expression of TNF-α was increased in the total brain of rats that received TTM at 33 °C after TGI.

### 2.4. TTM at 33 °C and 36 °C Attenuates Activation and Polarization of Microglia and Macrophages

We assessed the morphological changes in the brain and number of microglia/macrophages using specific markers for microglia at day 3 (72 h) after injury. Rats that underwent NT at 37.5 °C showed an enlarged ameboid phenotype Iba-1(ionized calcium binding adaptor molecule 1)-positive cells at 37.5 °C in the CA1 and PC, whereas these cells at 33 °C and 36 °C were mainly small with ramified cell bodies (Figure 4A–D and Figure 5A,B). Ramified microglial cells are composed of long branching processes and a small cellular body. These are commonly distributed at specific regions throughout the entire central nervous system in the absence of dying cells. TTM at 33 °C and 36 °C decreased the number of Iba-1-positive cells by approximately 72% to 86% compared to the corresponding numbers in the NT at 37.5 °C group. Thus, rats that received TTM at 33 °C and 36 °C did not significantly differ in assessments of microglial activity. We used ionized calcium binding adaptor molecule 1 (Iba-1) to detect microglia/macrophages in the CA1 and PC. In addition, to identify the polarized microglia/macrophages, they were labeled with cluster of differentiation 86 (CD86) against the M1 phenotype and cluster of differentiation 206 (CD206) against the M2 phenotype. The numbers of each labeled microglia/macrophages were also measured (Figure 4E,F and Figure 5C,D). Compared with the increased numbers of CD86- and CD206-positive cells with Iba-1 in rats under NT at 37.5 °C, TTM at 33 °C and 36 °C reduced the number of these positive cells by approximately 90% (Figure 4E,F) and 91% (Figure 5C,D), respectively. However, the number of CD86- and CD206-positive cells was not significantly different between TTM at 33 °C and 36 °C. TTM at 33 °C and 36 °C also showed that the ratio of CD206- to CD86-positive cells was one-fold and there was no significant difference between them. Thus, we found that there was no effect on the increase of the M2 phenotype of microglia in TTM at 33 °C and 36 °C.

### 2.5. TTM at 33 °C and 36 °C Modulated Cytokine Production in the TGI

To identify specific cytokines and compare the expressions of cytokines according to core temperatures, we conducted the cytokine array with whole brain tissue at 24 h after TGI (Figure 6A–D). We found that the expressions of several cytokines increased in the brain following TGI. In comparison with NT at 37.5 °C, TTM at 33 °C and 36 °C significantly reduced the expression levels of several cytokines, such as fractalkine/CX3CL1, intracellular adhesion molecule-1 (sICAM-1/CD54), and tissue inhibitor of metalloproteinases 1 (Figure 6E–G). Thus, we found that TTM at 33 °C and 36 °C regulated the levels of inflammation-related genes and proteins. Thus, TTM after TGI at 33 °C and 36 °C equally modulated the levels of sterile inflammation-related genes and proteins.

## 3. Discussion

In this study, we established a rat model for TTM at 36 °C and 33 °C after TGI, which induces ischemic/reperfusion injury of the total brain, similar to the one that occurs with the return of spontaneous circulation after sudden cardiac arrest (CA). We found that TTM at both 33 °C and 36 °C showed significant protective effects against neuronal cell apoptosis compared to normothermia of 37.5 °C. Our results also indicate that TTM at the higher temperature of 36 °C had neuroprotective effects equivalent to TTM at 33 °C. We confirmed by FJB that at 24 h post-injury, TTM at 36 °C and 33 °C intuitively attenuated neuronal cell death in cornu ammonis 1 (CA1) with significant improvements in behavioral function that lasted up to 72 h. There were similar neuroprotective effects between TTM at 36 °C and 33 °C.

TTM has been known to affect two main pathways (intrinsic and extrinsic) that lead to apoptotic cell death [5]. The intrinsic pathway acts within the cell at the level of the mitochondria by modulating the release of cytochrome c, the initiation of caspase-8, and the expression of the B-cell lymphoma 2 (Bcl-2) family of proteins [5,10,21]. TTM decreases pro-apoptotic Bcl-2 family members, like BAX, and promotes anti-apoptotic Bcl-2 members [5,10]. The extrinsic pathway in ischemic brain injury is activated via cell surface first apoptosis signal (Fas) receptors binding with their ligand (FasL) to initiate apoptosis via caspase-8 activation [5,10,22]. TTM also suppresses the expression of both Fas and FasL as well as the initiation of caspase-8 due to decreased solubility of the FasL [5,10]. Compared with 37.5 °C, we found that TTM at 33 °C and 36 °C provided significant protective effects against apoptotic cell death via decreased expressions of BAX and caspase-8. TTM at 33 °C and 36 °C equally diminished the expressions against BAX and caspase-8 in apoptotic cell death.

Recent studies have shown that it is mainly inflammation that contributes to neurological disorders after ischemic brain injury. HMGB1 is actively and passively released as a cytokine inflammation mediator of the two major pathways during injury causing organ damage [11]. Cellular damage occurs by passive pathways. In contrast, when immunologically competent cells such as monocytes, macrophages, and microglia are exposed to pathogen-associated molecular patterns, microbe-associated molecular patterns, and inflammatory mediators, including TNF-α, IL-1, and interferon gamma, the active pathway slows the secretion of HMGB1 by intracellular signal transduction with extracellular products [11]. In the vicious cycle of inflammatory responses after sterile injury, HMGB1 is significantly associated with the activation and polarization of microglia [4,19,20]. Our study revealed that TTM at 36 °C and 33 °C equally decreased the extracellular release of HMGB1 in the CA1 and PC in the early stages after TGI, in addition to the decreased mRNA expression of pro-inflammatory mediators, such as TNF-α, IL-1β, and IL-6, throughout the brain for 24 h after TGI. Based on these findings, both TTM at 36 °C and 33 °C help to prevent the spread of damage in the ischemic region after TGI. In one exception of our study, we found an increase in the mRNA expression of TNF-α in the group of rats that had received TTM at 33 °C after TGI. Since TTM at 33 °C requires extreme changes in temperature, we caution that TTM at 33 °C may cause adverse effects during the induction or rewarming stages (see below). As mentioned above, HMGB1 can be released passively from the apoptotic cells or actively by monocytes after exposure to apoptotic cell bodies during apoptosis [11]. TTM can interrupt the brutal cycle of inflammatory damage that occurs after TGI.

In the acute phase of an ischemic stroke, several cytokines are closely associated with the induction of cell-death cascades. In the central nervous system, the chemokine fractalkine/CX3CL1 is known to regulate communication between neurons and microglia. Previous studies report that fractalkine/CX3CL1 deficiency inhibited stroke-induced cell death, especially apoptotic cell death [23,24]. Upregulation of intracellular adhesion molecule 1 disturbed the blood–brain barrier structure, leading to extravasation of inflammatory cells and subsequent neuroinflammatory reactions [25,26]. In addition, tissue inhibition of metalloproteinases 1 is known to be related to excitotoxicity and inflammatory cascades [27,28]. To limit TGI-induced cell death by increasing inflammatory cytokines, we conducted paradigms of TTM after TGI. Consequently, we found that TTMs at 36 °C and 33 °C decreased the levels of these cytokines and can assert that sustained hypothermic conditions may reduce the brain damage associated with the release of various inflammatory cytokines (Figure 3).

Microglia play a critical role in brain inflammation [29]. Microglial activation is an early response to brain ischemia [30]. Although microglia exist in an extremely ramified form in their resting state, upon activation after ischemic injury, these cells often undergo morphologic changes to an amoeboid form [31]. We found that TTMs suppress this process in the CA1 and PC after TGI. In addition, contradictory functions of microglia reflect distinct phenotypes of cells produced under different microenvironments [18,30]. Activated microglia of the M1 type are the classic form and release destructive proinflammatory mediators. By contrast, activated microglia of M2 are the alternative form and execute clearance of cellular debris through phagocytosis while releasing factors for protection and repairs [18]. In previous studies of brain injury, the application of hypothermia diminished microglial activation after 3 days and induced microglial polarization toward the beneficial M2 phenotype [17,32]. Our study demonstrated that TTM at both 36 °C and 33 °C significantly suppressed M1 activation at 3 days post-TGI in comparison with TTM at 37.5 °C. However, there was no evidence that both TTM at 36 °C and 33 °C promoted microglial polarization toward the beneficial M2 phenotype. However, we propose that the lack of activated cell polarization toward M1 and M2 phenotype did not occur because TTM effectively blocks the activation of microglia. Consequently, TTM mitigates microglial activation due to ischemic injury. In clinical situations, the application of TTM cannot guarantee complete neurological recovery in patients with the return of spontaneous circulation after CA.

Recent studies provide novel therapeutic perspectives to central nervous system remodeling by expanding knowledge of microglia and macrophage phenotypes [18,33]. The new target of this approach is to promote switching phenotype activation toward beneficial M2 or preventing the activation of the proinflammatory and detrimental M1 phenotype, by using inducers or inhibitors of key switch molecules [18,30,33]. Ischemic brain injury is effectively reduced by phenotype regulation through supplementation of M2 inducers such as interleukin 4 [18,33]. Lentiviral delivery of M2-stimulating factors like interleukin 10 may promote M2 polarization and lessen the switch from the M2 to M1 in the inflammatory microenvironment [18,33].

Therefore, the promotion of M2 could be considered a new adjunctive neuroprotective therapy in CA. Further studies are needed to verify the neuroprotective effect of TTM against microglial activation and polarization in patients with the return of spontaneous circulation after cardiac arrest.

Despite the variability by the undetermined depth of core temperature, unlike patients with ischemic stroke and traumatic brain injury who remain awake and breathe spontaneously, unconscious survivors of cardiac arrest can tolerate lower and longer cooling because of airway protection, ventilation, appropriate sedation, and shivering control; hence, it may not be critical to determine the proper core temperature between TTM at 33 °C and 36 °C. However, TTM at 36 °C may be beneficial for patients after CA for several reasons. First, hypothermia is associated with various side effects. As shivering is a major adverse effect of TTM, aggressive control against shivering is essential to maximize the effect of TTM. Peripheral vasoconstriction is induced at 36.5 °C and shivering is triggered at 35.5 °C in a healthy human. Shivering decreases the delivery of oxygen to brain tissue and increases metabolic demands. TTM at 36 °C can decrease the risk of brain damage caused by the development of shivering more than TTM at 35 °C [9,34,35]. In addition, the detrimental increases in intracranial pressure and brain edema can cause injuries during the TTM phase of rewarming to normal core temperature [9,35]. As TTM at 36 °C can minimize the rewarming time between the target and normal temperatures, it can also maximize the neuroprotective effects by decreasing rewarming injuries.

This study has a few limitations that should be addressed. First, the results from an animal study cannot be directly extrapolated to human patients because of the considerable interspecies variability between animal models and human patients in cerebral vascular anatomy and in the immune response to transient global ischemic injury. Second, based on the results of several clinical trials, the 2015 Cardiopulmonary Resuscitation and Emergency Cardiovascular Care guidelines recommend the earliest time to prognosticate a poor neurologic outcome is 72 h after cardiac arrest. We selected 72 h as observation period. In the clinical setting, it is possible to clearly distinguish between good and bad neurological outcomes at 72 h after cardiac arrest. There is no significant difference in neurological outcome identified at 72 h after cardiac arrest over time. Further studies are required to validate long term neurologic outcomes and changes of inflammatory cascades against TTM.

## 4. Materials and Methods

### 4.1. Ethical Approval and Experimental Animals

Experimental procedures on all animals were performed in accordance with the guidelines of the Institutional Animal Care and Use Committee of the College of Medicine, Hallym University (Protocol # Hallym R1 [2018-34]) and the National Institutes of Health guidelines. In the current study, all experiments used healthy male Sprague Dawley rats (weight range 300 to 320 g; aged 8 weeks; DBL Co., Chungcheongbuk-do, Eumseong-gun, Korea). All rats were housed in a consistently maintained animal care room (temperature: 22 ± 2 °C, humidity: 55 ± 5%, light turn on/off cycle: 12 h interval) and given standard feed (Purina, Gyeonggi-do, Korea) and fresh water ad libitum. During and throughout experimental procedures, we used 2–3% isoflurane anesthesia to minimize any suffering.

### 4.2. Experimental Animal Model for Transient Global Ischemia

We used an experimental animal disease model of TGI in this study. Kawai et al. reported that the transient global ischemia animal disease model was associated with cardiac arrest [36]. To begin our thorough investigation, Sprague Dawley rats were deeply anesthetized with 2 to 3% isoflurane. Isoflurane gas was ventilated using a mix of 70% nitrous oxide and 30% oxygen. Body temperature was consistently checked using a homeothermic monitoring system (Harvard Apparatus, Holliston, MA, USA). To monitor hypotension and consistently check systemic arterial blood pressure, a cannula was inserted into the femoral artery. After cannulation, common carotid arteries were located on the bilateral sides of the trachea muscle and carefully isolated using a surgical microscope (SZ61, Olympus, Shinjuku, Japan) to avoid the stimulation of the vagus nerve and to enhance dissection accuracy. After isolation, to continuously monitor the changes in electroencephalography, electrodes were placed into the bilateral burr holes. In order to induce a transient ischemic condition and adjust the systemic arterial blood pressure, blood was drained from the femoral artery through the inserted cannula (target range of 50 mmHg systolic and 40 mmHg diastolic). Next, we isolated the bilateral common carotid arteries and occluded those using surgical clamps (Fine Science Tools, Foster City, CA, USA). Once occluded, we checked the electroencephalography waveforms. At the isoelectric point, conditions were maintained for 7 min. Following unclamping, the previously blocked blood circulation to the brain was restored by reperfusion of the drained blood (Figure 7).

### 4.3. Targeted Temperature Management

Rats were randomly divided into six groups: Sham at 37.5 °C (Sham + 37.5 °C), Sham at 36 °C (Sham + 36 °C), Sham at 33 °C (Sham + 33 °C), normal temperature (NT) at 37.5 °C (TGI + 37.5 °C), TTM at 33 °C (TGI + 33 °C), and TTM at 36 °C (TGI + 36 °C). To induce hypothermic conditions (target core temperatures: 33 ± 0.5 °C, 36 ± 0.5 °C, 37.5 ± 0.5 °C), the body of the experimental animal was covered with packed ice. The core temperatures were carefully monitored. Target temperature was reached within approximately 30 min and was maintained for 4 h (for HMGB1 analysis and cytokine array) or 3 h (for the adhesive removal test and all the other biochemical and histological analyses) using the feedback-controlled heating pad and surface cooling with ice packs. (1) To determine the effects of TTM after transient global ischemia throughout the 72-h experiment, we conducted daily behavioral assessment and histological evaluations, 3 h TTM and 1 h rewarming process was required for our study. (2) To observe HMGB1 and cytokine release after global ischemic insult within the short time window immediately after insult, we conducted TTM for 4 h after transient global ischemia and animals were immediately sacrificed after termination of 4 h TTM. Before the initiation of TTM, vecuronium bromide (0.9 mg/kg) was administrated intramuscularly to avoid shivering. After termination of hypothermia, the rewarming process was conducted using the heating pad (rewarming speed: 0.5 °C/10 min; targeted rewarming temperature: 37 ± 0.5 °C) for 1 h (Figure 7A–E).

### 4.4. Adhesive Removal Test

To test whether hypothermic conditions attenuated sensorimotor impairments after TGI induction, an adhesive removal test was conducted for 3 consecutive days after TGI. The entire process of the adhesive removal test was performed in accordance with previously reported methods [37,38,39]. This test consisted of several steps. The rats were acclimated to the transparent testing cage (size: 45 × 35 × 20 cm) for 1 to 2 min. Meanwhile, we prepared two pieces of adhesive tape (1 cm × 1 cm). Once acclimated, the two pieces of tape were attached to the pad of each forepaw. Immediately after the tape was attached, the removal time was checked. The time it took for the rat to detach the adhesive tape using its mouth or by shaking its paws was recorded (maximum time 120 s). During the test period, in cases where the adhesive tape could not be detached, the removal time was considered as the maximum time of 120 s. This process was repeated for five consecutive trials with several minutes between each trial.

### 4.5. Detection of Degenerating Neurons

To evaluate neuronal death after TGI, brain sections (30 µM) were put on silane-coated slides (MUTO PURE CHEMICALS CO., LTD, Hongo Bunkyo-ku, Tokyo, Japan). To detect degenerating neurons, brain sections were stained with the Fluoro-Jade B (FJB; Millipore (Burlington, MA, USA)) staining method. We performed deparaffinization with paraffin-embedded brain sections of 4 μm. Next, we washed the brain tissue with distilled water for 1 min followed by deep immersion in 0.06% potassium permanganate (Sigma-Aldrich, Saint Louis, MO, USA) solution for 20 min. After 20 min, we rinsed them in distilled water for 2 min. They were then reacted in 0.001% FJB for 30 min and rinsed 3 times for 1 min each in distilled water. After rinsing, the slides were dried to 50 °C by gentle air flow (SW-90D, Sang Woo scientific Co., Bucheon, Korea), dehydrated in xylene, and mounted with ProLong™ Diamond Antifade Mountant with 4,6-diamidino 2-phenylindole (DAPI; Invitrogen., Carlsbad, CA, USA). To verify TGI-induced neuronal death, the samples were observed with a fluorescence microscope using blue (450–490 m) wavelengths. We used approximately 6 to 8 coronal brain sections that were excised from each animal, initiated 4.0 mm posterior from the bregma. To correctly count the number of FJB-positive cells, we used a blind observer. They counted the FJB-positive cells in the hippocampal cornus ammonis 1 regions (CA1) of the bilateral hemispheres. The total number of FJB-positive cells from each hippocampal region was used for statistical analysis.

### 4.6. TUNEL Assay

To verify whether TTMs prevent TGI-induced apoptotic cell damage, TUNEL assays were conducted using a DeadEnd™ Fluorometric TUNEL system (Promega, Madison, MI, USA) in accordance with the manufacturer’s instructions. The stained sections were evaluated using a confocal microscope (LSM700, Carl Zeiss, Oberkochen, Germany). The number of TUNEL-positive cells in the CA1 or parietal cortex (PC) was normalized to the number of cells counted from the sham brains.

### 4.7. Immunofluorescence

Using a microtome (Leica, Wetzlar, Germany), brain sections were cut from a region of infarction at a thickness of 4 μm. The sections were permeabilized with 0.1% Triton X-100 buffer and blocked with 3% goat serum (or 2.5% horse serum) in phosphate-buffered saline for 1 h at room temperature. The samples were then incubated with the primary antibody overnight. The antibodies used for the immunofluorescence analyses were: rabbit anti-Iba-1 (1:200, Wako Pure Chemical Industries, Osaka, Japan); mouse anti-CD86 (1:200, LSBio, Seattle, WA, USA); mouse anti-CD206 (1:50, Santa Cruz Biotechnology Inc., USA); rabbit anti-caspase 8 (1:100, Abcam, Cambridge, UK); mouse anti-Bcl-2-associated X (BAX) (1:100, Novusbio, Littleton, CO, USA); and rabbit anti-HMGB1 (1:100, Abcam, UK).

The samples were washed and incubated for 2 h at 37 °C with secondary fluorescent antibodies conjugated to Goat anti-rabbit IgG Alexa-fluor 488 (1:200; Invitrogen, Carlsbad, CA, USA) and mouse IgGκ binding protein-PE (1:50; Santa Cruz Biotechnology Inc., Dallas, TX, USA). Finally, the samples were mounted with ProLong™ Diamond Antifade Mountant with DAPI (Invitrogen, USA) and a confocal microscope (LSM 700, Carl Zeiss GmbH, Germany) was used to observe the stained sections.

### 4.8. Cytokine Array

The hemispheric tissue of the rat brains was homogenized in phosphate-buffered saline with protease inhibitor (Bertin technologies Montigny, Montigny-le-Bretonneux, France). Protein quantitation was conducted using the protocol included in the Pierce BCA protein assay kit. Proteome Profiler array was conducted using the Cytokine Array Panel A (R&D systems, Minneapolis, MN, USA) in accordance with the manufacturer’s instructions. Blots were visualized and evaluated by the ECLTM Western Blotting Analysis System (GE Healthcare, Chicago, IL, USA) and imaged using a LAS 4000 mini image analyzer (Fujifilm, Tokyo, Japan). The blots were quantized by HLImage++ (Western Vision software) to analyze this array.

### 4.9. Real Time Reverse Transcription Polymerase Chain Reaction

Tissue RNA was extracted using the Hybrid-R kit (305-010, GeneAll^®^ Biotechnology, Seoul, Korea). Complimentary DNAs were obtained from 500 ng of total RNA using the PrimeScript™ 1st strand cDNA Synthesis Kit (Takara Bio, Kusatsu, Japan). PrimerQuest (Integrated DNA Technologies, Coralville, IA, USA) was used to design the primers for interleukin 1 beta (IL-1β), interleukin 6 (IL-6), tumor necrosis factor-α (TNF-α), and glyceraldehyde 3-phosphate dehydrogenase (GAPDH). Quantitative polymerase chain reaction was performed using a 7500 ABI system (Applied Biosystems, Foster, CA, USA) with SYBR-Green reagent (GenDEPOT). The results were as follows:

TNF-α Forward: TAG CAA ACC ACC AAG CAG AGTNF-α Reverse: AGA GAA CCT GGG AGT AGA TAA GGIL-1β Forward: CTA TGG CAA CTG TCC CTG AAIL-1β Reverse: GGC TTG GAA GCA ATC CTT AAT CIL-6 Forward: GAA GTT AGA GTC ACA GAA GGA GTGIL-6 Reverse: GTT TGC CGA GTA GAC CTC ATA GGAPDH Forward: ACT CCC ATT CTT CCA CCT TTGGAPDH Reverse: CCC TGT TGC TGT AGC CAT ATT

### 4.10. Statistical Analysis

All experiments were randomized, and all analyses were conducted by investigators blinded to the experimental conditions. Statistical analyses were performed using standard statistical methods (*t* test, Systat Software, Inc., San Jose, CA, USA). The significance level was presented as the mean ± the SEM. Differences between groups were analyzed using one-way ANOVA followed by Bonferroni’s post hoc test using GraphPad Prism software for windows. *p* < 0.05 was considered statistically significant.

## 5. Conclusions

Our study suggests that TTM at 33 °C and 36 °C produce equivalent neuroprotective effects by reducing apoptosis and activation of microglia, as well as neuroinflammation, by inhibiting the release of inflammatory precursors, such as HMGB1 and inflammatory cytokines. Moreover, TTM at 33 °C and 36 °C effectively and equivalently suppressed the activation of M1 phenotypes. However, there is no direct link between TTM and the promotion of M2 after TGI. This study provides new mechanistic insights into TTM at 36 °C, and these implications may help to establish a more robust theoretical framework for clinical applications of TTM at 36 °C in TGI.

## Figures and Tables

**Figure 1 ijms-22-05114-f001:**
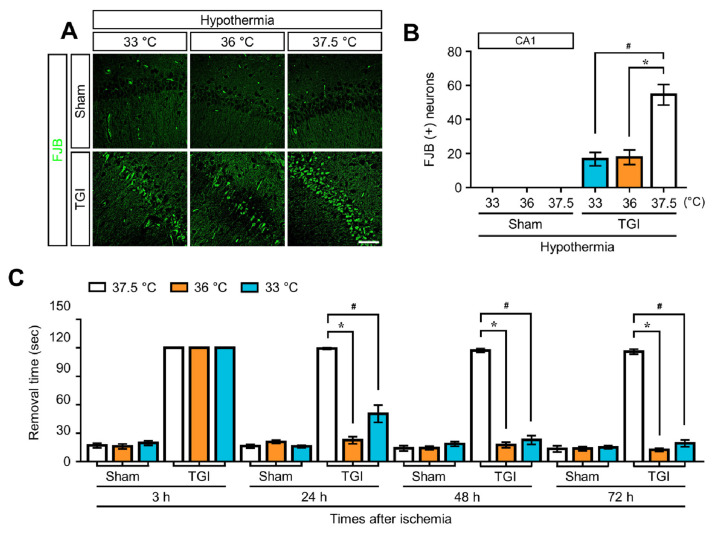
Target temperature management (TTM) suppresses transient global ischemia (TGI)-induced neuronal death and sensorimotor deficit. (**A**) Brain sections were stained with Fluoro-Jade B fluorescent dye to detect degenerating neurons. TGI at a normal temperature of 37.5 °C significantly increased the number of fluorescent cells in the hippocampal cornus ammonis 1 region. These cells were significantly decreased after TGI-TTM at 33 °C and 36 °C. There were no differences between the sham groups. Scale bar = 50 μm. (**B**) Bar graph represents the total number of Fluoro-Jade B-positive cells (one way-ANOVA, multiple comparison following Bonferroni adjustment. Sham + 33 °C, *n* = 5; Sham + 36 °C, *n* = 5; Sham + 37.5 °C, *n* = 5; TGI + 33 °C, *n* = 8; TGI + 36 °C, *n* = 5; TGI + 37.5, *n* = 8. * *p* and # *p* < 0.05). (**C**) Adhesive removal test to observe sensorimotor deficit proceeded 3 to 72 h after TGI (tested daily). There were no differences in the removal time in each sham group assessment. TGI-TTM at 33 °C and 36 °C significantly reduced the removal time (time-dependent) compared to TGI at normal temperature at 37.5 °C (one way-ANOVA, multiple comparison following Bonferroni adjustment. Sham + 33 °C, *n* = 5; Sham + 36 °C, *n* = 5; Sham + 37.5 °C, *n* = 5; TGI + 33 °C, *n* = 9; TGI + 36 °C, *n* = 7; TGI + 37.5, *n* = 8. * *p* and # *p* < 0.05).

**Figure 2 ijms-22-05114-f002:**
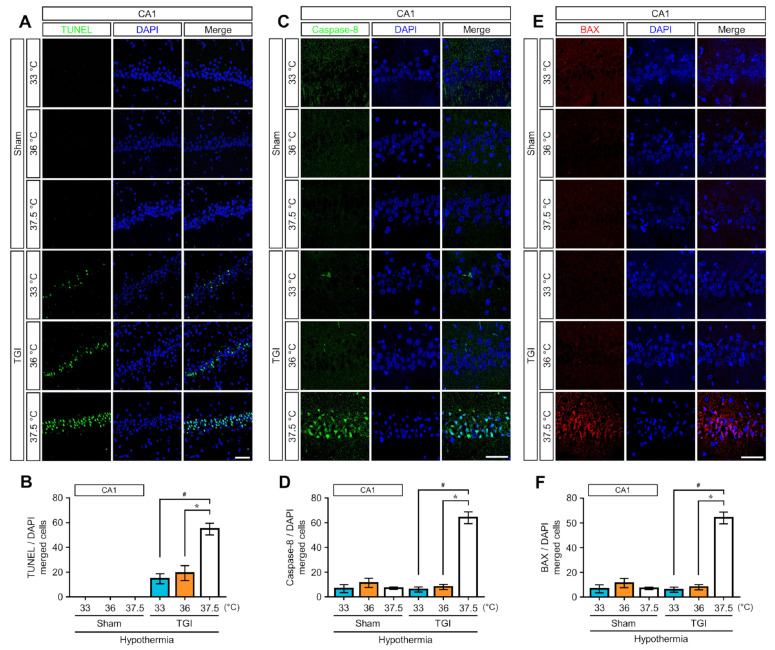
TTM suppresses apoptotic cell death 72 h after TGI. (**A**) Representative histological images display the number of TUNEL-positive cells in the cornus ammonis 1 of the sham or the hippocampus and in TGI. There were no differences in the number of TUNEL-positive cells in each sham group. TGI at a normal temperature at 37.5 °C resulted in an increase in the number of these cells at 72 h. TGI-TTM at 33 °C and 36 °C significantly decreased TUNEL-positive cells compared to TGI at normal temperature at 37.5 °C. Scale bar = 50 μm. (**B**) Representative bar graphs show the number of TUNEL-positive cells in the sham and TGI groups (one way-ANOVA, multiple comparison following Bonferroni adjustment. Sham + 33 °C, *n* = 5; Sham + 36 °C, *n* = 5; Sham + 37.5 °C, *n* = 5; TGI + 33 °C, *n* = 8; TGI + 36 °C, *n* = 5; TGI + 37.5, *n* = 8. * *p* and # *p* < 0.05). TTM at 33 °C and 36 °C reduces apoptosis-associated caspase-8 and Bcl-2-associated protein (BAX) expression at 72 h after TGI. The expression of apoptotic cell death-related caspase-8 and pro-apoptotic member, BAX, was suppressed by TTM at 33 °C and 36 °C after TGI. (**C**) In the sham groups, there were no differences in caspase-8 expression. TGI-normal temperature (NT) at 37.5 °C upregulated the expression of caspase-8 within the hippocampal cornus ammonis 1 region. TTM at 33 °C and 36 °C significantly suppressed caspase-8 expression. Scale bar = 50 μm. (**E**) The apoptosis-related substrate, BAX, was overexpressed in TGI-NT 37.5 °C. In the sham groups, there were no differences in BAX expression. TGI-TTM at 33 °C and 36 °C significantly reduced BAX expression. Scale bar = 50 μm. (**D**) Bar graph indicates the percentage of caspase-8 expression. Caspase-8 expression is greater in TGI-NT at 37.5 °C than at 33 °C and 36 °C (one way-ANOVA, multiple comparison following Bonferroni adjustment. Sham + 33 °C, *n* = 3; Sham + 36 °C, *n* = 3; Sham + 37.5 °C, *n* = 3; TGI + 33 °C, *n* = 3; TGI + 36 °C, *n* = 3; TGI + 37.5, *n* = 4. * *p* and # *p* < 0.05). (**F**) Graph shows that BAX expression decreased in both hypothermic paradigms (one way-ANOVA, multiple comparison following Bonferroni adjustment. Sham + 33 °C, *n* = 3; Sham + 36 °C, *n* = 3; Sham + 37.5 °C, *n* = 3; TGI + 33 °C, *n* = 3; TGI + 36 °C, *n* = 3; TGI + 37.5, *n* = 4. * *p* and # *p* < 0.05).

**Figure 3 ijms-22-05114-f003:**
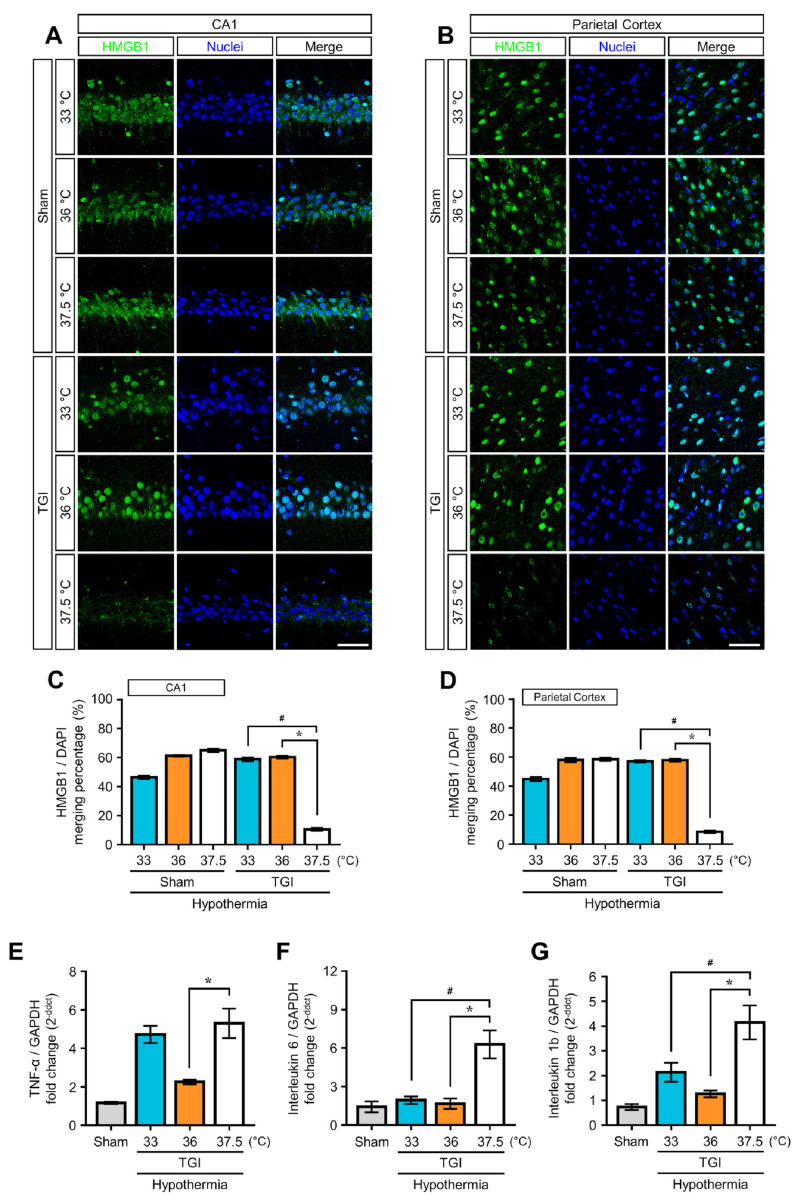
TTM blocks high mobility group box 1 (HMGB1) and cytokines release after TGI. HMGB1 level measured at 4 h after TGI. (**A**,**B**) Representative images showed co-localization of HMGB1 with nuclei (DAPI fluorescent stain). In the sham groups, HMGB1 expression was correlated with nuclei. TGI at normal temperature (37.5 °C) accelerated HMGB1 release from the nuclei, and TGI-TTM at 33 °C and 36 °C prevented HMGB1 release in the hippocampal cornus ammonis 1 region. Scale bar = 50 μm. (**C**,**D**) Percentage of HMGB1 and nuclei (DAPI) correlation in both hippocampal cornus ammonis 1 and parietal cortex in both hypothermic paradigms (one way-ANOVA, multiple comparison following Bonferroni adjustment. Sham + 33 °C, *n* = 4; Sham + 36 °C, *n* = 4; Sham + 37.5 °C, *n* = 4; TGI + 33 °C, *n* = 4; TGI + 36 °C, *n* = 4; TGI + 37.5, *n* = 4. * *p* and # *p* < 0.05). HMGB1 release from nuclei triggered pro-inflammatory cytokine expression. (**E**–**G**) Quantification of fold change (2^−ddct^ value) by quantitative polymerase chain reaction of tumor necrosis factor-alpha, interleukin-1β, and interleukin-6 versus glyceraldehyde 3-phosphate dehydrogenase. The expression of tumor necrosis factor-alpha, interleukin-1β, and interleukin-6 was decreased after TTM at 33 °C and 36 °C compared to the expression at 37.5 °C (normal temperature). TGI-TTM at 36 °C decreased tumor necrosis factor-alpha expression significantly more than TGI at 37.5 °C (one way-ANOVA, multiple comparison following Bonferroni adjustment. Sham, *n* = 3; TGI + 33 °C, *n* = 3; TGI + 36 °C, *n* = 3; TGI + 37.5, *n* = 3. * *p* and # *p* < 0.05).

**Figure 4 ijms-22-05114-f004:**
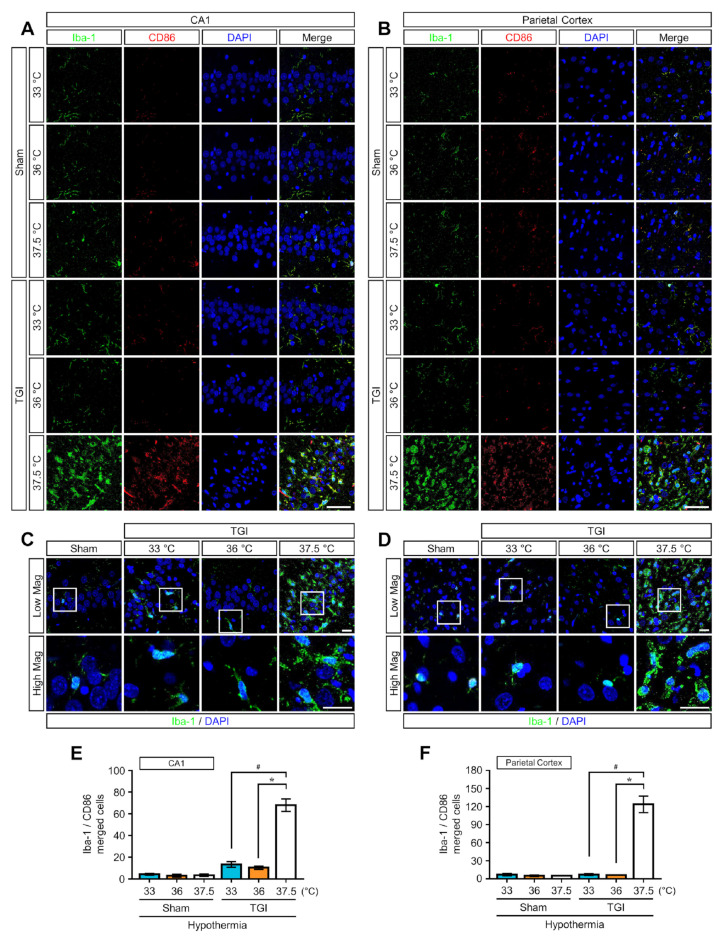
TGI-induced high mobility group box 1 release in the early stage triggers microglial activation and M1 type polarization. TTM at 33 °C and 36 °C inhibits this. (**A**,**B**) At 72 h after TGI, representative images show microglial activation and M1 type polarization in both the sham and TGI groups in the hippocampal cornus ammonis 1 and parietal cortex. The sham groups had no differences in microglial activation and polarization. TGI to normal temperature at 37.5 °C triggered microglial activation (Iba-1, green signal) and M1 type polarization (CD86, red signal). Scale bar = 50 μm. (**C**,**D**) Image shows low and high magnification of Iba-1-positive microglial cells in hippocampal cornus ammonis 1 region and parietal cortex. Scale bar = 20 μm. (**E**,**F**) Bar graph shows the percentage of Iba-1 and CD86 emerging in the hippocampal cornus ammonis 1 and parietal cortex. TGI-TTM at 33 °C and 36 °C had a lower percentage than TGI at 37.5 °C (one way-ANOVA, multiple comparison following Bonferroni adjustment. Sham + 33 °C, *n* = 3; Sham + 36 °C, *n* = 4; Sham + 37.5 °C, *n* = 6; TGI + 33 °C, *n* = 5; TGI + 36 °C, *n* = 8; TGI + 37.5, *n* = 6. * *p* and # *p* < 0.05).

**Figure 5 ijms-22-05114-f005:**
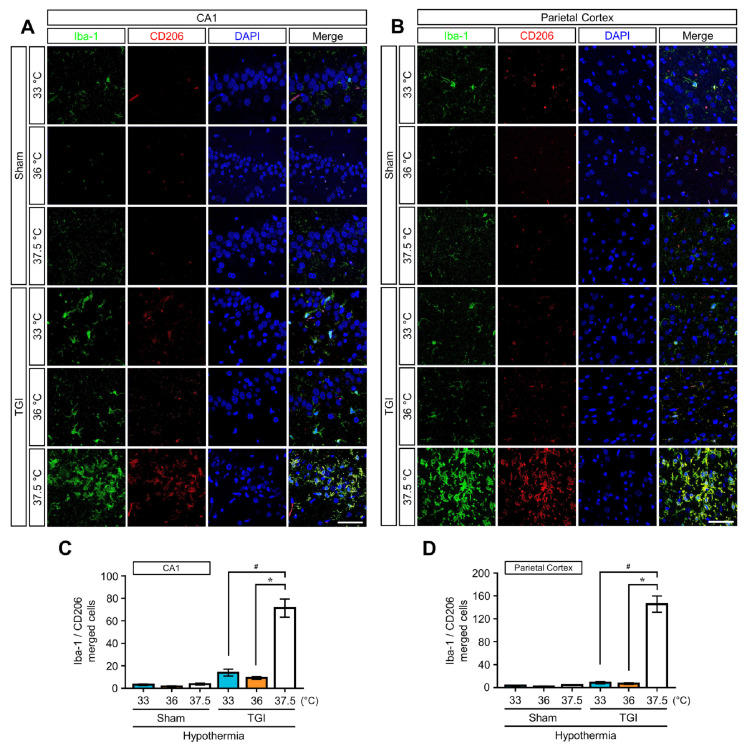
TTM at 33 °C and 36 °C suppresses changes in M2 phenotype microglia at 72 h after TGI. (**A**,**B**) Fluorescent imaging shows total microglia (Iba-1, green signal), the M2 type microglia (CD206, red signal), and merger with nuclei in the hippocampal cornus ammonis and parietal cortex regions. Scale bar = 50 μm. (**C**,**D**) Percentage of Iba-1 and CD206 merging at the hippocampal cornus ammonis and parietal cortex regions. Iba-1 and CD206 co-localization percentage was lower in TGI-TTM at 33 °C and 36 °C than in TGI at 37.5 °C (one way-ANOVA, multiple comparison following Bonferroni adjustment. Sham + 33 °C, *n* = 3; Sham + 36 °C, *n* = 4; Sham + 37.5 °C, *n* = 6; TGI + 33 °C, *n* = 5; TGI + 36 °C, *n* = 8; TGI + 37.5, *n* = 6. * *p* and # *p* < 0.05).

**Figure 6 ijms-22-05114-f006:**
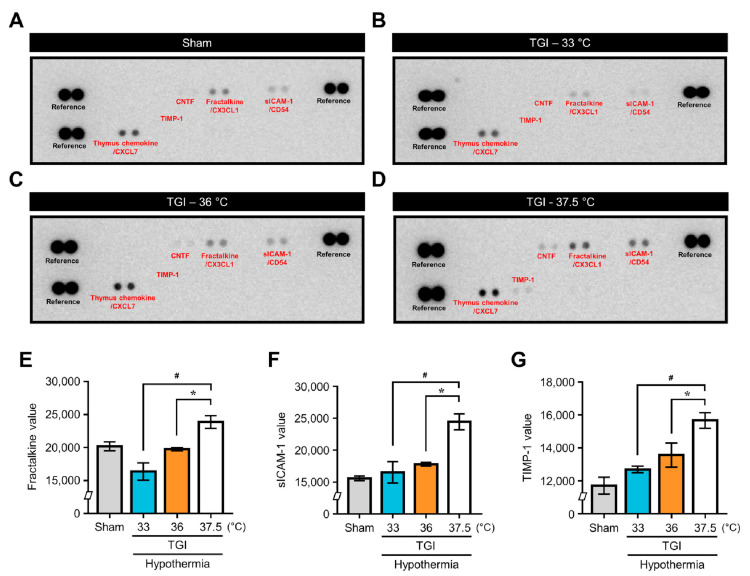
Cytokine array screening of the normal temperature and hypothermic paradigms of TGI. (**A**–**D**) Representative chemiluminescent images show the proteome profiler array at normal temperature and TTM conditions at 24 h after TGI. Selected cytokines involving ciliary neurotrophic factor, fractalkine/CX3CL1 (chemokine ligand 1), slCAM-1/CD54, thymus chemokine/CXCL7 (chemokine ligand 7), and tissue inhibitor of metalloproteinases 1 are indicated in the box, and compared with sham or TGI to normal temperature at 37.5 °C, TTM at 33 °C and 36 °C. (**E**–**G**) Analyzed value of the selected cytokines (fractalkine/CX3CL1, slCAM-1/CD54, tissue inhibitor of metalloproteinases 1) displayed by a representative bar graph. Expression of these cytokines was remarkably attenuated by TTM at 33 °C and 36 °C after TGI (one way-ANOVA, multiple comparison following Bonferroni adjustment. Sham, *n* = 3; TGI + 33 °C, *n* = 3; TGI + 36 °C, *n* = 3; TGI + 37.5, *n* = 3. * *p* and # *p* < 0.05).

**Figure 7 ijms-22-05114-f007:**
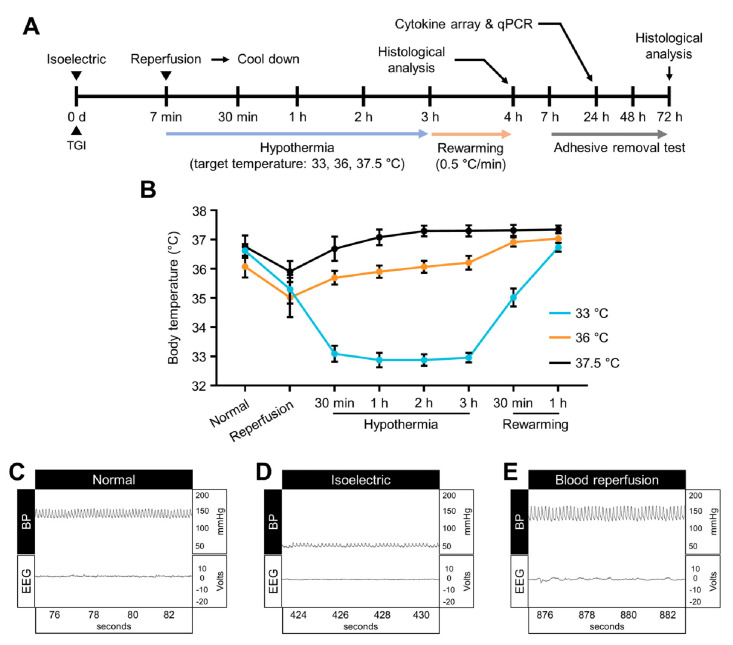
Experimental design for targeted temperature management (TTM). (**A**) Timeline representing the entire experimental design following TTM. After termination of TTM, several analyses were conducted to verify the effects of TTM. (**B**) Changes of systemic body temperature during TTM. (**C**–**E**) Arterial blood pressure and electroencephalograph pattern during transient global ischemia. Before initiation of the set isoelectric point; blood withdraw range 40 (diastolic) to 50 (systolic) mmHg. Blood pressure and electroencephalography findings were completely restored after blood reperfusion. Targeted temperature management (TTM) at 33 °C and 36 °C decreases hippocampal neuronal death and sensorimotor deficits.

## Data Availability

The datasets generated during and/or analyzed during the current study are available from the corresponding authors on reasonable request.

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
