# Peer review of "Transient Global Ischemia-Induced Brain Inflammatory Cascades Attenuated by Targeted Temperature Management"

_ijms, 2021, doi:10.3390/ijms22105114_

Round 1

Reviewer 1 Report

This is a comprehensive study in rats investigating the effect of cooling on transient global ischemia-induced inflammation and damage to the brain. Cooling (TTM) with either 33 °C or 36 °C was compared to normal temperature. Both TTM groups showed attenuation of neuronal cell death and apoptosis and inflammatory response. Behavioral function after 72 hours was improved after TTM. The chosen methods seem very appropriate and cover a lot of different aspects. The topic is of clinical interest.

Major Criticism

  1. The authors conclude that TTM at 36 °C should be considered safe and effective after cardiac arrest. The present study does in no way support this strong clinical conclusion. The study is performed in rats and does not necessarily translate to humans. The conclusion should be rewritten and should only take the present results into account.
  2. The last paragraph in the introduction states the results. This is not the proper place in the manuscript to do that. I will suggest that the last paragraph is used for the exact aim of the study and corresponding hypotheses instead.
  3. The experimental design contains a no flow period of 7 minutes. The TTM is started in the relevant two groups immediately after the no flow period. In the clinical setting cooling will not be commenced that quickly after a cardiac arrest. Why did the authors choose not to have a certain period of “no touch” before cooling to mimic the clinical situation? Could the immediate cooling actually have enhanced the effect of cooling on the different parameters? This should be discussed.
  4. The reason for choosing examination of degeneration of the neurons specific in the hippocampus should be explained. Why not other areas of the brain?
  5. In figure 1 G and H the color code of the different temperature groups are confusing and probably incorrect. Should 37.5 be white or black?
  6. An observation period of 72 hours has been chosen. Thus the changes and differences seen must be categorized as short term. Cerebral recovery in the clinical setting often occurs over weeks or months. Is it possible that longer follow up would have shown other results? This should be discussed
  7. A paragraph in the Discussion with study limitations in general is missing and could add to the value of the paper.
  8. The last paragraph in the discussion is used for discussion of clinical aspects of TTM 33 and 36. Some of the statements are rather speculative, and the authors should be more careful about the conclusion in this regard. It would be more appropriate to refer to published clinical randomized trials and perhaps also ongoing studies examining different temperature levels. I would prefer that the authors end the discussion with a paragraph with a short conclusion focusing on the exact findings in the present study and the meaning of these.
  9. The number of mice in the different groups should be stated in the manuscript. I could not find it.
  10. The statistics include parametric methods. I would imagine that the number of mice in the different groups was small, and non-parametric statistics would therefore have been more appropriate. Was the data checked for normal distribution before statistical analysis?

Minor criticism

  1. The sentence in the abstract stating that: “we performed experiment in rat model” needs to be rephrased with a better English grammar: we performed experiments in a rat model or we performed an experiment in a rat model are possible suggestions from me as a non-English college.

Author Response

Dear Dr. Andrea Popa, Assigned Editor, International Journal of Molecular Science

Manuscript ID: ijms- 1192726

Title: Transient Global Ischemia-induced Brain Inflammatory Cascades Attenuated by Targeted Temperature Management

I appreciate the opportunity to revise this manuscript. I respect the reviewers’ helpful comments and have responded to both reviewers’ comments point-by-point, revised as indicated below <yellow highlights>. I hope this revised manuscript is acceptable for publication in your journal.

Reviewer(s)' Comments to Author:

This is a comprehensive study in rats investigating the effect of cooling on transient global ischemia-induced inflammation and damage to the brain. Cooling (TTM) with either 33 °C or 36 °C was compared to normal temperature. Both TTM groups showed attenuation of neuronal cell death and apoptosis and inflammatory response. Behavioral function after 72 hours was improved after TTM. The chosen methods seem very appropriate and cover a lot of different aspects. The topic is of clinical interest.

Major Criticism

  1. The authors conclude that TTM at 36 °C should be considered safe and effective after cardiac arrest. The present study does in no way support this strong clinical conclusion. The study is performed in rats and does not necessarily translate to humans. The conclusion should be rewritten and should only take the present results into account.

<Response: Thank you for your comment; we agree with you. Accordingly, we have revised this point to the conclusion session of manuscript. We suggest that TTM at 36°C and 33°C can be an excellent neuroprotective treatment for protecting neuronal death via attenuation of inflammatory cascades after TGI. This study provides new mechanistic insights into TTM at 36°C and these implications may help to establish a more robust theoretical framework for neuroprotection of TTM at 36 °C in TGI.>

  1. The last paragraph in the introduction states the results. This is not the proper place in the manuscript to do that. I will suggest that the last paragraph is used for the exact aim of the study and corresponding hypotheses instead.

<Response: Thank you for helpful comment. We re-wrote last paragraph in the introduction. “Thus, in our study, we compared the neuroprotective effects of TTM at 33°C and 36°C after CA using a transient global ischemia (TGI) model. We hypothesized that TTM at both 33°C and at 36°C may produce a robust, neuroprotective effect on ischemic injuries of the whole brain by preventing the release of HMGB1 from damaged cells, thereby inhibiting apoptosis, microglial activation, and imbalanced microglial polarization, as well as consequently diminishing the propagation of the inflammatory response.”>

  1. The experimental design contains a no flow period of 7 minutes. The TTM is started in the relevant two groups immediately after the no flow period. In the clinical setting cooling will not be commenced that quickly after a cardiac arrest. Why did the authors choose not to have a certain period of “no touch” before cooling to mimic the clinical situation? Could the immediate cooling actually have enhanced the effect of cooling on the different parameters? This should be discussed.

<Response: Thank you for helpful comment. We agree with reviewer’s concern. We added it in introduction and discussion part of manuscript. There are reasons of not choose “no touch” period. First, animal study cannot be extrapolated to clinical studies because considerable interspecific variations between animal disease model and clinical patients. Second, we focused that HMGB1 was excessively released from damaged cells within a short time [1]. To demonstrate whether hypothermia can block HMGB1 releasing within early time phase after transient global ischemia, immediately cooling was performed.>

  1. The reason for choosing examination of degeneration of the neurons specific in the hippocampus should be explained. Why not other areas of the brain?

<Response: Thank you for meaningful comment. Especially global ischemic brain injury, hippocampal region has selective vulnerability [2]. One of hippocampal region cornu ammonis 1 (CA1) was closely related to ischemia-induced cell death [3,4]. Zhang and Kovalenko et al founded that prolonged global ischemic insult kills numerous hippocampal neurons within several days [5,6]. Based on previous studies, we targeting hippocampus for determine global ischemia-induced neuronal loss.>

  1. In figure 1 G and H the color code of the different temperature groups is confusing and probably incorrect. Should 37.5 be white or black?

<Response: Thank you for helpful comment. We re-made figure 1 H color code to avoid confusing individual groups. All bar graph of different temperature groups has black outline, and 37.5°C be white color, 36°C be orange color and 33°C be blue color.>

  1. An observation period of 72 hours has been chosen. Thus the changes and differences seen must be categorized as short term. Cerebral recovery in the clinical setting often occurs over weeks or months. Is it possible that longer follow up would have shown other results? This should be discussed

<Response: Thank you for your comment. We agree with your comment. Accordingly, we have revised this point to the manuscript. On the basis of the results of several clinical trials, the 2015 Cardiopulmonary Resuscitation and Emergency Cardiovascular Care guidelines recommend the earliest time to prognosticate a poor neurologic outcome is 72 hours after cardiac arrest [7]. We selected 72 hours as observation period. This time until prognostication can be even longer than 72 hours after cardiac arrest if the residual effect of sedation or paralysis confounds the clinical examination. In clinical setting, it is possible to clearly distinguish between good and bad neurological outcomes at 72 hours after cardiac arrest. There is no significant difference in neurological outcome identified at 72 hours after cardiac arrest over time. We think that it is difficult to show other results in longer follow up. However, we have revised this point to the limitation of manuscript. Further studies are required to validate long term neurologic outcomes and changes of inflammatory cascades against TTM.>

  1. A paragraph in the Discussion with study limitations in general is missing and could add to the value of the paper.

<Response: Thank you for helpful comment. We added a new paragraph in the discussion with study limitations. “This study has some limitations. First, the results from animal study cannot be extrapolated to human patients because the considerable interspecies variability between animal models and human patients in cerebral vascular anatomy and in the immune response to transient global ischemic injury. Second, on the basis of the results of several clinical trials, the 2015 Cardiopulmonary Resuscitation and Emergency Cardiovascular Care guidelines recommend the earliest time to prognosticate a poor neurologic outcome is 72 hours after cardiac arrest. We selected 72 hours as observation period. In clinical setting, it is possible to clearly distinguish between good and bad neurological outcomes at 72 hours after cardiac arrest. There is no significant difference in neurological outcome identified at 72 hours after cardiac arrest over time. Further studies are required to validate long term neurologic outcomes and changes of inflammatory cascades against TTM.”>

  1. The last paragraph in the discussion is used for discussion of clinical aspects of TTM 33 and 36. Some of the statements are rather speculative, and the authors should be more careful about the conclusion in this regard. It would be more appropriate to refer to published clinical randomized trials and perhaps also ongoing studies examining different temperature levels. I would prefer that the authors end the discussion with a paragraph with a short conclusion focusing on the exact findings in the present study and the meaning of these.

<Response: Thank you for your comment. We agree with your comment. Accordingly, we have deleted this point from the manuscript>

  1. The number of mice in the different groups should be stated in the manuscript. I could not find it.

<Response: Thank you for reviewer’s comment. We stated number of rat in the different groups in figure legend of manuscript.>

  1. The statistics include parametric methods. I would imagine that the number of mice in the different groups was small, and non-parametric statistics would therefore have been more appropriate. Was the data checked for normal distribution before statistical analysis?

<Response: Thank you for reviewer’s comment. In data analysis, we discussed this point with our statistician (Hye Sung Lee, PhD, Department of Research Affairs, Biostatistics Collaboration Unit, Yonsei University College of Medicine). First, we comprehensively checked the normality based on three methods: Shapiro-Wilk test, histogram, and Q-Q plot and identified trends of normal distribution. In addition, the non-parametric method is a method to calculate the statistic and p-value based on rank regardless of the observed value. In the small sample size, if the ranks are the same, the same statistic and p-value are calculated. So our statistician recommended our study should be analyzed using the parametric method. We analyzed our data using the parametric method.>

Minor Criticism

  1. The sentence in the abstract stating that: “we performed experiment in rat model” needs to be rephrased with a better English grammar: we performed experiments in a rat model or we performed an experiment in a rat model are possible suggestions from me as a non-English college.

<Response: Thank you for meaningful comment. We re-wrote it “we performed experiment in rat model” to “we performed an experiment in a rat model”.>

References

  1. Kim, J.B.; Sig Choi, J.; Yu, Y.M.; Nam, K.; Piao, C.S.; Kim, S.W.; Lee, M.H.; Han, P.L.; Park, J.S.; Lee, J.K. HMGB1, a novel cytokine-like mediator linking acute neuronal death and delayed neuroinflammation in the postischemic brain. The Journal of neuroscience : the official journal of the Society for Neuroscience 2006, 26, 6413-6421, doi:10.1523/jneurosci.3815-05.2006.
  2. Nikonenko, A.G.; Radenovic, L.; Andjus, P.R.; Skibo, G.G. Structural features of ischemic damage in the hippocampus. Anat Rec (Hoboken) 2009, 292, 1914-1921, doi:10.1002/ar.20969.
  3. Hsu, M.; Sik, A.; Gallyas, F.; Horvath, Z.; Buzsaki, G. Short-term and long-term changes in the postischemic hippocampus. Ann N Y Acad Sci 1994, 743, 121-139; discussion 139-140, doi:10.1111/j.1749-6632.1994.tb55790.x.
  4. Zhan, R.Z.; Nadler, J.V.; Schwartz-Bloom, R.D. Depressed responses to applied and synaptically-released GABA in CA1 pyramidal cells, but not in CA1 interneurons, after transient forebrain ischemia. Journal of cerebral blood flow and metabolism : official journal of the International Society of Cerebral Blood Flow and Metabolism 2006, 26, 112-124, doi:10.1038/sj.jcbfm.9600171.
  5. Kovalenko, T.; Osadchenko, I.; Nikonenko, A.; Lushnikova, I.; Voronin, K.; Nikonenko, I.; Muller, D.; Skibo, G. Ischemia-induced modifications in hippocampal CA1 stratum radiatum excitatory synapses. Hippocampus 2006, 16, 814-825, doi:10.1002/hipo.20211.
  6. Zhang, F.; Liu, C.L.; Hu, B.R. Irreversible aggregation of protein synthesis machinery after focal brain ischemia. J Neurochem 2006, 98, 102-112, doi:10.1111/j.1471-4159.2006.03838.x.
  7. Callaway, C.W.; Donnino, M.W.; Fink, E.L.; Geocadin, R.G.; Golan, E.; Kern, K.B.; Leary, M.; Meurer, W.J.; Peberdy, M.A.; Thompson, T.M.; et al. Part 8: Post-Cardiac Arrest Care: 2015 American Heart Association Guidelines Update for Cardiopulmonary Resuscitation and Emergency Cardiovascular Care. Circulation 2015, 132, S465-482, doi:10.1161/cir.0000000000000262.

Reviewer 2 Report

The paper presented by the authors shows the effects of targeted temperature management in an experimental model of cardiac arrest in the brain. The study is well conducted with clear and relevant results.  However, the manuscript should be further improved with some minor changes. Also, some details of the study need to be better defined.

  1. Maybe for a layout mistake material and methods chapter is the 4th chapter and is inserted in the text after discussion. Please reorganize the text as expected with material and methods after the introduction chapter.

  1. The last sentence of the introduction (lines 82-88) reports the conclusions of the study. The authors should modify the last sentence and report only the purpose of the study and not the results and conclusions.

  1. As specified in lines 50-51, the Targeted Temperature after cardiac arrest is maintained “ at least 24 hours”, in these experiments, TTM is maintained for 3 or 4 hours. Why have authors chosen this timing? Why in the experiments there is two different timing of TTM? These choices should be explained in material and methods.

  1. Authors should specify the ethical committee approval for the experiments and the experimental protocol register number.

  1. In Figure 1A-B-C-D-E, the experiment timeline, the temperature maintained during the experiment and the EEG and arterial pressure pattern are shown. These five figures could be inserted in the material and methods chapter and separated from Figure 1 FGH that shows the ischemia effects with FJB. If there is a problem with a limited number of figures, authors could merge Figure 1 F-G-H with figure 2 that shows apoptosis and neuronal death.

  1. The authors should specify why they chose to observe the rats for three days; an intermediate time point could be helpful?

  1. In the material and methods chapter, lines 366-383, authors should insert a citation for the model of cardiac arrest with hemorrhagic shock.

Author Response

Dear Dr. Andrea Popa, Assigned Editor, International Journal of Molecular Science

Manuscript ID: ijms- 1192726

Title: Transient Global Ischemia-induced Brain Inflammatory Cascades Attenuated by Targeted Temperature Management

I appreciate the opportunity to revise this manuscript. I respect the reviewers’ helpful comments and have responded to both reviewers’ comments point-by-point, revised as indicated below <yellow highlights>. I hope this revised manuscript is acceptable for publication in your journal.

Reviewer(s)' Comments to Author:

The paper presented by the authors shows the effects of targeted temperature management in an experimental model of cardiac arrest in the brain. The study is well conducted with clear and relevant results. However, the manuscript should be further improved with some minor changes. Also, some details of the study need to be better defined.

  1. Maybe for a layout mistake material and methods chapter is the 4th chapter and is inserted in the text after discussion. Please reorganize the text as expected with material and methods after the introduction chapter.

<Response: Thank you for reviewer’s comment. In the template for IJMS, the layout order is following ‘introduction’ (1th), ‘results’ (2nd), ‘discussion’ (3rd) and ‘material and method’ (4th). We arranged it in order.>

  1. The last sentence of the introduction (lines 82-88) reports the conclusions of the study. The authors should modify the last sentence and report only the purpose of the study and not the results and conclusions.

<Response: Thank you for helpful comment. We re-wrote last paragraph in the introduction. “Thus, in our study, we compared the neuroprotective effects of TTM at 33°C and 36°C after CA using a transient global ischemia (TGI) model. We hypothesized that TTM at both 33°C and at 36°C may produce a robust, neuroprotective effect on ischemic injuries of the whole brain by preventing the release of HMGB1 from damaged cells, thereby inhibiting apoptosis, microglial activation, and imbalanced microglial polarization, as well as consequently diminishing the propagation of inflammatory responses.”>

  1. As specified in lines 50-51, the Targeted Temperature after cardiac arrest is maintained “ at least 24 hours”, in these experiments, TTM is maintained for 3 or 4 hours. Why have authors chosen this timing? Why in the experiments there is two different timing of TTM? These choices should be explained in material and methods.

<Response: Thank you for meaningful comment. We explained it in material and method of manuscript. Cardiopulmonary resuscitation and emergency cardiovascular care guideline recommended clinically application of therapeutic hypothermia time maintained at least 24 hours. Previous studies have demonstrated that maintaining 2~4 hours of hypothermia rescues brain damage against cerebral diseases [1-3]. In this study, we divided TTM maintaining time 3 hours and 4 hours. “(1) To determine effects of TTM after transient global ischemia throughout the 72 hours experiment that daily conducted behavior assessment and histological evaluations, 3 hours TTM and 1 hour rewarming process was needed our study. (2) HMGB1 is rapidly released from damaged cells within a few hours [4]. To observe HMGB1 and cytokines releasing without risk of rewarming process from global ischemic insult within a short time, we conducted TTM for 4 hours after transient global ischemia, and animals were immediately sacrificed after termination of 4 hours TTM.”>

  1. Authors should specify the ethical committee approval for the experiments and the experimental protocol register number.

<Response: Thank you for reviewer’s comment. We added it in 4.1. Ethical Approval and Experimental Animals of manuscript. “Experimental procedures on all animals were performed in accordance with the guidelines of the Institutional Animal Care and Use Committee of the College of Medicine, Hallym University (Protocol # Hallym R1 [2018-34]) and the National Institutes of Health guidelines.”>

  1. In Figure 1A-B-C-D-E, the experiment timeline, the temperature maintained during the experiment and the EEG and arterial pressure pattern are shown. These five figures could be inserted in the material and methods chapter and separated from Figure 1 FGH that shows the ischemia effects with FJB. If there is a problem with a limited number of figures, authors could merge Figure 1 F-G-H with figure 2 that shows apoptosis and neuronal death.

<Response: Thank you for the helpful comment. We separated and re-arranged Figure 1. Figure 1A-B-C-D-E moved into material and method.>

  1. The authors should specify why they chose to observe the rats for three days; an intermediate time point could be helpful?

<Response: Thank you for your comment. We agree with your comment. On the basis of the results of several clinical trials, the 2015 Cardiopulmonary Resuscitation and Emergency Cardiovascular Care guidelines proposed the selection and maintenance of a constant target temperature between 32 and 36 °C for at least 24 h in TTM. The guidelines recommend the earliest time to prognosticate a poor neurologic outcome is 72 hours after cardiac arrest. We selected 72 hours as observation period. In clinical setting, it is possible to clearly distinguish between good and bad neurological outcomes at 72 hours after cardiac arrest. There is no significant difference in neurological outcome identified at 72 hours after cardiac arrest over time. Therefore, we conducted several histological outcomes for demonstrating the effects of TTM after three days following transient global ischemia. Further studies are required to validate long term neurologic outcomes and changes of inflammatory cascades against TTM. Accordingly, we have added this point to the limitations of manuscript.>

  1. In the material and methods chapter, lines 366-383, authors should insert a citation for the model of cardiac arrest with hemorrhagic shock.

<Response: Thank you for reviewer’s comment. We added it in material and methods. “Kawai et al. reported that transient global ischemia animal disease model associated with cardiac arrest [5]>

References

  1. Kim, K.; Jo, Y.H.; Rhee, J.E.; Kim, T.Y.; Lee, J.H.; Lee, J.H.; Kwon, W.Y.; Suh, G.J.; Lee, C.C.; Singer, A.J. Effect of speed of rewarming and administration of anti-inflammatory or anti-oxidant agents on acute lung injury in an intestinal ischemia model treated with therapeutic hypothermia. Resuscitation 2010, 81, 100-105, doi:10.1016/j.resuscitation.2009.09.020.
  2. Lee, J.H.; Kim, K.; Jo, Y.H.; Lee, M.J.; Hwang, J.E.; Kim, M.A. Effect of valproic acid combined with therapeutic hypothermia on neurologic outcome in asphyxial cardiac arrest model of rats. Am J Emerg Med 2015, 33, 1773-1779, doi:10.1016/j.ajem.2015.08.036.
  3. Ahn, J.H.; Lee, T.K.; Tae, H.J.; Kim, B.; Sim, H.; Lee, J.C.; Kim, D.W.; Kim, Y.S.; Shin, M.C.; Park, Y.; et al. Neuronal Death in the CNS Autonomic Control Center Comes Very Early after Cardiac Arrest and Is Not Significantly Attenuated by Prompt Hypothermic Treatment in Rats. Cells 2021, 10, doi:10.3390/cells10010060.
  4. Lee, J.H.; Yoon, E.J.; Seo, J.; Kavoussi, A.; Chung, Y.E.; Chung, S.P.; Park, I.; Kim, C.H.; You, J.S. Hypothermia inhibits the propagation of acute ischemic injury by inhibiting HMGB1. Molecular brain 2016, 9, 81, doi:10.1186/s13041-016-0260-0.
  5. Kawai, K.; Nitecka, L.; Ruetzler, C.A.; Nagashima, G.; Joo, F.; Mies, G.; Nowak, T.S., Jr.; Saito, N.; Lohr, J.M.; Klatzo, I. Global cerebral ischemia associated with cardiac arrest in the rat: I. Dynamics of early neuronal changes. Journal of cerebral blood flow and metabolism : official journal of the International Society of Cerebral Blood Flow and Metabolism 1992, 12, 238-249, doi:10.1038/jcbfm.1992.34.

Author Response

Dear Dr. Andrea Popa, Assigned Editor, International Journal of Molecular Science

Manuscript ID: ijms- 1192726

Title: Transient Global Ischemia-induced Brain Inflammatory Cascades Attenuated by Targeted Temperature Management

I appreciate the opportunity to revise this manuscript. I respect the reviewers’ helpful comments and have responded to both reviewers’ comments point-by-point, revised as indicated below <yellow highlights>. I hope this revised manuscript is acceptable for publication in your journal.

Reviewer(s)' Comments to Author:

This is a well performed study and a manuscript that is generally well written. There are a number of times where abbreviations are left unidentified or identified well after their initial use. The are enumerated below. Attention to that issue as well as the additional comments below should make the manuscript better.

Major Comments

  1. Between lines 63-67, add: Following an ischemic event HMGB1 is translocated to the intracellular fluid (ICF) and secreted into the extracellular space. Studies have indicated that intracellular HMGB1 plays an important role in the regulation of energy homeostasis and transcription. In contrast, it has been reported that extracellular HMGB1 directs blood brain barrier breakdown, neuroimmune activities, and neuronal death (see Front Cell Neurosci. 2019 Apr 2;13:127).

<Response: Thank you for your comment; we agree with you. Accordingly, we have added this comment to the introduction of manuscript. “Following an ischemic event HMGB1 is translocated to the intracellular fluid (ICF) and secreted into the extracellular space. Studies have indicated that intracellular HMGB1 plays an important role in the regulation of energy homeostasis and transcription. In contrast, it has been reported that extracellular HMGB1 directs blood brain barrier breakdown, neuroimmune activities, and neuronal death [1].”>

  1. In figure 2, it would be useful to explain that 4′,6-diamidino-2-phenylindole (DAPI) is a blue-fluorescent DNA stain that exhibits ~20-fold enhancement of fluorescence upon binding to AT regions of double stranded DNA.

<Response: Thank you for helpful comment. We added it in results of manuscript. “These immunostainings correlated with 4′,6-diamidino-2-phenylindole (DAPI) that is a blue-fluorescent DNA stain, ~20-fold enhancement of fluorescence upon binding to AT regions of double stranded DNA.”>

  1. See lines 167-171: Please clarify the impact of differences in TNF-α on gene expression and TNF-α on mRNA expression with TTM at 33°C. Do they negate each other?

<Response: Thank you for reviewer’s comment. We re-wrote it. “gene expression of TNF-α” to “mRNA expression of TNF-α”>

  1. To support your findings, you may wish to note in the figure 3 legend that: Ramified microglial are composed of long branching processes and a small cellular body. These cells are commonly found at specific locations throughout the entire brain and spinal cord in the absence of dying cells.

<Response: Thank you for the helpful comment. We agree with the reviewer’s concern. Accordingly, we have added this comment to the results of manuscript (2.4. TTM at 33°C and 36°C Attenuates Activation and Polarization of Microglia and Macrophages).>

  1. The normal body temperature of the rat is about 37.5-37.8° C with a tendency to poikilothermia. Human body temperature can have a wide range, from 36.1°C to 37.2°C. It is well recognized that the body temperature of older men and women is lower than that of younger people. The median age of sudden cardias death victims in the US is between age 66 and 68. Hypothermia is defined as a body core temperature below 35.0 °C. On lines 351-352 you state “the application of TTM at 36°C is increasing worldwide”. If available, please provide more references/evidence that the target of TTM at 36°C can be extrapolated to humans.

<Response: The application of TTM at 36°C has been increasing worldwide [2]. However, until now, it is unknown if TTM at 36°C produces comparable neuroprotective effects against brain damage after CA with respect to molecular mechanisms. On the basis of the results of several clinical trials, the 2015 Cardiopulmonary Resuscitation and Emergency Cardiovascular Care guidelines proposed the selection and maintenance of a constant target temperature between 32 and 36 °C for at least 24 h in TTM. With respect to survival and neurological benefit, one randomized trial published in 2013 compared patients treated with TTM at either 33°C or 36°C and demonstrated that the effects of TTM at 36°C were just as beneficial as those at 33°C [3]. Recently, the application of TTM at 36°C is increasing worldwide. Several clinical trials included patients receiving TTM at 36 °C and there was no significant differences of mortality and neurologic outcomes between TTM at 36°C and 33°C. Despite the increasing application of TTM to clinicians, it can be difficult to extrapolated animal experiment results directly to humans. This may be due to the considerable interspecies variability animal stroke models and human patients show in cerebral vascular anatomy and in the immune response to ischemic injury [4,5].>

Minor Comments

  1. On line 13, the phrase “higher mobility rates” is confusing. Did you mean lower physical mobility rates or high mobility group box 1 release rates?

<Response: We re-wrote it (mobility → mortality)>

  1. On line 77, the words “In addition” are superfluous and should be deleted.

<Response: We deleted it>

  1. On line 94, identify CA1 as cornu ammonis 1 (CA1). This should occur before line 251.

<Response: We re-wrote it>

  1. On line 94, does the abbreviation PC refer to the precuneus, the posterior cingulate cortex, the Parietal Cortex, the Perirhinal Cortex, the Piriform Cortex, the Prefrontal Cortex, the procerebrum or another structure/anatomical region? Please clarify.

<Response: We re-wrote it. PC means parietal cortex.>

  1. On line 95, please write: “Fluoro-Jade B (FJB)”.

<Response: We re-wrote it>

  1. On line 96, does the abbreviation NT mean normothermia, normothermic, or nonthermal? Please clarify.

<Response: We added abbreviation of NT. It means normothermia.>

  1. On lines 101-102, the words: “…showed significant improvement (no statistical difference)…” are potentially confusing and appear contradictory. Please delete “(no statistical difference)”, which seems to be explained in the next sentence.

<Response: We deleted it>

  1. On line 118, do you mean each sham group assessment? Please clarify.

<Response: That means each sham group assessment. We re-wrote it.>

  1. On line 119, change: “to normal temperature at 37.5°C” to “at normal temperature of 37.5°C”.

<Response: We re-wrote it>

  1. On line 124, please identify TUNEL as terminal deoxynucleotidyl transferase dUTP nick end labeling (TUNEL) and BAX as BCL2 Associated X, Apoptosis Regulator (BAX). Please also note that caspace-8 plays a central role in the execution-phase of cell [HYPERLINK "https://en.wikipedia.org/wiki/Apoptosis" \o "Apoptosis"].

<Response: We re-wrote it>

  1. On line 195, change: “Iba1” to “Ionized calcium binding adaptor molecule 1 (Iba1)”.

<Response: We re-wrote it>

  1. On line 197, identify CD86 as Cluster of Differentiation 86 (CD86) and note that it can signal for self-regulation and cell-cell association, or for attenuation of regulation and cell-cell disassociation.

<Response: We re-wrote it>

  1. On line 197, identify CD286 as toll like receptor 6 (CD286 or TRL6) and note that it which plays a fundamental role in activation of innate immunity.

<Response: We re-wrote CD206 as cluster of differentiation 206 (CD206).>

  1. On line 230, please identify sICAM-1/CD54 as intercellular adhesion molecule-1(sICAM1/CD54).

<Response: We re-wrote it>

  1. On line 252, please identify Bcl 2 as B-cell lymphoma 2 (Bcl 2).

<Response: We re-wrote it>

  1. On line 259, please identify Fas as First apoptosis signal (Fas).

<Response: We re-wrote it>

  1. On lines 283-284, you write: “…caution that TTM at 33°C may cause adverse effects during the induction or rewarming stages”. This is not elaborated upon until lines 339- 349. It would be useful to change the statement to “…caution that TTM at 33°C may cause adverse effects during the induction or rewarming stages (see below)”.

<Response: We re-wrote it>

  1. On line 296, you write: “To preserve TGI-induced cell death…”. Did you mean “To limit TGI-induced cell death…”.

<Response: It means “To limit TGI-induced cell death…”. We re-wrote “To preserve” to “To limit”.>

  1. On line 304, change: “a process of morphologic changes” to either “a process of morphologic change” or “morphologic changes”.

<Response: We re-wrote “a process of morphologic changes” to “morphologic changes”.>

  1. On line 362, change: “During the throughout” to “During and throughout”.

<Response: We re-wrote it>

  1. On line 378, change: “range” to “target range”.

<Response: We re-wrote it>

  1. On line 391, change: “targeting” to “target”.

<Response: We re-wrote it>

  1. On line 401, you write: “To test whether hypothermic conditions maintaining after TGI-induced sensorimotor impairments, …”. Did you mean to write “To test whether hypothermic conditions attenuated sensorimotor impairments after TGI-induction”?

<Response: Thank you for reviewer’s comment. To clarify, we re-wrote it “To test whether hypothermic conditions attenuated sensorimotor impairments after TGI-induction”.>

  1. On line 452, change: “37oC” to “37°C”.

<Response: We re-wrote it>

  1. It is traditional to follow companies’ names with their location. Usually both are included in parentheses after the technologies’ is named. You do this nicely on lines 470-471 by writing “…Hybrid-R kit (305-010; GeneAll® Biotechnology, 470 Seoul, Korea)”.

<Response: We re-wrote it>

References

  1. Ye, Y.; Zeng, Z.; Jin, T.; Zhang, H.; Xiong, X.; Gu, L. The Role of High Mobility Group Box 1 in Ischemic Stroke. Frontiers in cellular neuroscience 2019, 13, 127, doi:10.3389/fncel.2019.00127.
  2. Johnson, N.J.; Danielson, K.R.; Counts, C.R.; Ruark, K.; Scruggs, S.; Hough, C.L.; Maynard, C.; Sayre, M.R.; Carlbom, D.J. Targeted Temperature Management at 33 Versus 36 Degrees: A Retrospective Cohort Study. Crit Care Med 2020, 48, 362-369, doi:10.1097/CCM.0000000000004159.
  3. Callaway, C.W.; Coppler, P.J.; Faro, J.; Puyana, J.S.; Solanki, P.; Dezfulian, C.; Doshi, A.A.; Elmer, J.; Frisch, A.; Guyette, F.X.; et al. Association of Initial Illness Severity and Outcomes After Cardiac Arrest With Targeted Temperature Management at 36 degrees C or 33 degrees C. JAMA Netw Open 2020, 3, e208215, doi:10.1001/jamanetworkopen.2020.8215.
  4. Sommer, C.J. Ischemic stroke: experimental models and reality. Acta Neuropathol 2017, 133, 245-261, doi:10.1007/s00401-017-1667-0.
  5. Dirnagl, U. Bench to bedside: the quest for quality in experimental stroke research. Journal of cerebral blood flow and metabolism : official journal of the International Society of Cerebral Blood Flow and Metabolism 2006, 26, 1465-1478, doi:10.1038/sj.jcbfm.9600298.

Round 2

Author Response

Dear Dr. Andrea Popa, Assigned Editor, International Journal of Molecular Science

Manuscript ID: ijms- 1192726

Title: Transient Global Ischemia-induced Brain Inflammatory Cascades Attenuated by Targeted Temperature Management

I appreciate the opportunity to revise this manuscript. I respect the reviewers’ helpful comments and have responded to both reviewers’ comments point-by-point, revised as indicated below <yellow highlights>. I hope this revised manuscript is acceptable for publication in your journal.

Reviewer(s)' Comments to Author:

The manuscript is improved. Attention to a few additional items should complete it.

Major Comments

  1. On line 377-378, change: “…the detrimental increase of intracranial pressure and brain edema can cause injuries associated with the TTM phase of rewarming to normal core temperature” to “…increases in intracranial pressure and brain edema can cause injuries during the TTM phase of rewarming to normal core temperature”.

<Response: We re-wrote it.>

  1. Lines 191-193, you discuss TNF-α in rats after TTM at 33°C. On lines 313-314 you write: “…we found an increase in the mRNA expression of TNF-α in the total brain of a rat that had received TTM at 33°C after TGI”. These statements are inconsistent and confusing. Do you mean one rat or the group of rats that received TTM at 33°C after TGI?

<Response: We appreciate reviewer’s comment. That means the group of rats received TTM at 33°C after TGI. To clarify, we re-wrote it.>

  1. On lines 289-290, you write: “…promotes anti-apoptotic members like Bcl-2”. Do you mean “…promotes anti-apoptotic Bcl-2 members”?

<Response: Thank you for reviewer’s comment. That means promotes anti-apoptotic Bcl-2 members. We re-wrote it.>

  1. On lines 348-350, change: “However, we propose that, based on the absolute lack of activated cell polarization toward M1 and M2 phenotype did not occur because TTM effectively blocks the activation of microglia” to “However, we propose that, lack of activated cell polarization toward M1 or M2 phenotype did not occur because TTM effectively blocks the activation of microglia”.

<Response: We re-wrote it.>

  1. On lines 397-398, you write: “There is no significant difference in neurological outcome identified at 72 hours after cardiac arrest over time”. Do you mean no significant additional neurological recovery is likely after 72 hours?

<Response: Thank you for your comment. We agree with your comment. On the basis of the results of several clinical trials [1], the 2015 Cardiopulmonary Resuscitation and Emergency Cardiovascular Care guidelines proposed the selection and maintenance of a constant target temperature between 32 and 36 °C for at least 24 h in TTM. The guidelines recommend the earliest time to prognosticate a poor neurologic outcome is 72 hours after cardiac arrest. We selected 72 hours as observation period. In clinical setting, it is possible to clearly distinguish between good and bad neurological outcomes at 72 hours after cardiac arrest. There is no significant difference in neurological outcome identified at 72 hours after cardiac arrest over time. Therefore, we conducted several histological outcomes for demonstrating the effects of TTM after three days following transient global ischemia. Further studies are required to validate long term neurologic outcomes and changes of inflammatory cascades against TTM. Accordingly, we have added this point to the limitations of manuscript.>

Longstreth et al demonstrated that of the 279 patients who awakened, 25% did so by time of admission, 71% by 1 day after their out-of-hospital cardiac arrest, 86% by 2 days, and 92% by 3 days. Patients who never awakened died during the study period; survival was 50% at 3.5 days, 30% at 1 week and 4% at 1 month [1].

In general, if circulation recovers after cardiac arrest and there is no consciousness rescue 72 hours later, prognosis is considered bad. In cases of recovered consciousness, almost patients were rescued at 72 hours after cardiac arrest. In contrast, no recovered consciousness 72 hours later, 50% patients were died at 3.5 days. There has been no further studies since this study, and in the clinical cases have observed 72 hours as an important prognosis time. CPC (cerebral performance category) grade 1, 2 is good outcome and 3, 4, 5 is poor outcome. If there was no abnormal phenotype within 72 hours, it considered that good outcome.

Minor Comments

 On line 13, change: “the 36° ” to “36° ”.

<Response: We re-wrote it.>

  1. On line 20, delete: “intuitively”.

<Response: We deleted it.>

  1. On line 25, add the words “In addition” prior to “TTM at 36°C ameliorates”.

<Response: We re-wrote it.>

  1. On line 32, change: “higher” to “a high”

<Response: We re-wrote it.>

  1. On line 103, change: “the fluorescent dye” to “a fluorescent dye”.

<Response: We re-wrote it.>

  1. On lines 73-74: Is extracellular HMGB1 a danger signal or a dangerous trigger?

<Response: Extracellular-released HMGB1 is a dangerous trigger. It binds toll like receptors (TLRs), and HMGB1-bounded TLRs activate inflammatory cascades [2,3].>

  1. On line 372, delete: “mild”.

<Response: We deleted it.>

  1. On line 374, change: “The shivering” to “Shivering”.

<Response: We re-wrote it.>

  1. On line 457, change: “was needed our study” to either “was used in our study” or “was required for our study”. Please choose which phrase fits best.

<Response: We re-wrote it. “was needed our study” to “was required for our study”.>

References

  1. Longstreth, W.T., Jr.; Inui, T.S.; Cobb, L.A.; Copass, M.K. Neurologic recovery after out-of-hospital cardiac arrest. Ann Intern Med 1983, 98, 588-592, doi:10.7326/0003-4819-98-5-588.
  2. Das, N.; Dewan, V.; Grace, P.M.; Gunn, R.J.; Tamura, R.; Tzarum, N.; Watkins, L.R.; Wilson, I.A.; Yin, H. HMGB1 Activates Proinflammatory Signaling via TLR5 Leading to Allodynia. Cell Rep 2016, 17, 1128-1140, doi:10.1016/j.celrep.2016.09.076.
  3. Bertheloot, D.; Latz, E. HMGB1, IL-1alpha, IL-33 and S100 proteins: dual-function alarmins. Cell Mol Immunol 2017, 14, 43-64, doi:10.1038/cmi.2016.34.
